# Challenges of Embedding Fiber Bragg Grating Sensors in Castable Material: Influence of Material Shrinkage and Fiber Coatings on Ultrasonic Measurements

**DOI:** 10.3390/s25092657

**Published:** 2025-04-23

**Authors:** Nicolas Derrien, Maximilien Lehujeur, Xavier Chapeleau, Olivier Durand, Antoine Gallet, Nicolas Roussel, Béatrice Yven, Odile Abraham

**Affiliations:** 1Géophysique et Evaluation Non Destructive (GéoEND) Laboratory, Géotechnique, Environnement, Risques Naturels et Sciences de la Terre (GERS) Department, Université Gustave Eiffel, Nantes Campus, F-44344 Bouguenais, France; maximilien.lehujeur@univ-eiffel.fr (M.L.); olivier.durand@univ-eiffel.fr (O.D.); odile.abraham@univ-eiffel.fr (O.A.); 2Structure et Instrumentation Intégrée (SII) Laboratory, Inference for Structures (I4S) Team (Inria), Composants et Systèmes (COSYS) Department, Université Gustave Eiffel, Nantes Campus, F-44344 Bouguenais, France; xavier.chapeleau@univ-eiffel.fr; 3Laboratoire d’Intégration des Systèmes et des Technologies (LIST), Commissariat à l’Énergie Atomique et aux Énergies Alternatives (CEA), Université Paris-Saclay, Paris-Saclay Campus, F-91120 Palaiseau, France; antoine.gallet@cea.fr (A.G.); nicolas.roussel@cea.fr (N.R.); 4Agence Nationale Pour la Gestion des Déchets Radioactifs (ANDRA), F-92298 Châtenay-Malabry, France; beatrice.yven@andra.fr

**Keywords:** ultrasonics, optical fibers, bulk waves, polyurethane resins, acrylate coating, polyimide coating

## Abstract

Fiber optic sensors are increasingly used to measure dynamic strain fields caused by the propagation of mechanical waves. Their low intrusiveness when embedded within a structure makes them suitable for a wide range of applications. In this paper, the feasibility of integrating fiber Bragg gratings (FBGs) into castable materials for ultrasonic applications is investigated. We employed castable polyurethane resins, which are widely used in industry due to their reproducible and durable mechanical properties. Our study began with an analysis of fiber integration by examining the 1D strain profiles of two polyurethane resins during their polymerization and also the impact of their hardening on the central wavelength value of several FBGs spectra. Subsequently, we assessed the sensitivity of FBGs to ultrasonic waves generated at 100 kHz after resin polymerization. Specifically, we explored how the fiber coating influences the rate of energy transfer from the host material to the fiber core. Our findings demonstrate that the central wavelength shift in the FBG reflectivity spectra, caused by shrinkage during resin polymerization, can reach up to 10 nm. This shift must be considered when selecting FBG wavelengths to prevent the reflectivity spectra from falling outside the permissible range of the interrogation system. We measured exploitable ultrasonic waves propagating in the resin samples. Preliminary observations suggest the presence of early arrivals, which could potentially correspond to crosstalk effects between the FBGs even though they are centered at different wavelengths. Furthermore, we show that in dynamic strain fields caused by ultrasonic wave propagation, both acrylate and polyimide coatings transmit similar amounts of energy to the fiber core. These preliminary results highlight the potential of using FBGs as ultrasonic wave sensors embedded in castable materials such as polyurethane resins.

## 1. Introduction

Thanks to their light weight and low intrusiveness, Optical Fiber Sensors (OFSs) have attracted interest interest in Structural Health Monitoring (SHM). One of the major challenges with OFSs in SHM is measuring the dynamic strain caused by an ultrasonic wave. Two major technologies are currently used to measure dynamic strain fields with OFSs: Distributed Acoustic Sensing (DAS) and fiber Bragg gratings (FBGs). DAS technology, primarily used in geophysics, still faces challenges when used as an ultrasonic sensor. This is due to the integration of fiber deformations over a large gauge length (typically several meters) and a limited sampling frequency. On the contrary, FBG technology [1], although not distributed, provides an effective solution because it offers a sufficiently small sensing element (from few micrometers to several centimeters) that is sensitive to wavelengths in the ultrasonic frequency range [2,3,4,5,6,7]. Moreover, as this sensor is directly inscribed in an optical fiber, its diameter is very small (usually 125 µm with the core and the cladding) compared with other ultrasonic sensors. This enables it to be embedded within a structure with limited impact on its mechanical properties [8]. For this reason, the insertion of optical fibers used as sensors in various materials is of great technical interest, especially in SHM. In particular, embedding optical fibers in castable materials is an approach that allows the fiber to be confined within the material while ensuring optimal coupling with it.

In this study, we used polyurethane resins, whose mechanical properties are highly reproducible and temporally stable. However, during the casting process, there is a risk of breakage due to the internal shrinkage induced by the polymerization phase. To minimize this effect, the optical fiber is generally surrounded by one or several protective coatings. There are generally two types of protective coatings employed to protect optical fibers: acrylate and polyimide. For static strain measurements, it has been shown that these protective coatings play a role in the sensitivity of the fiber to deformation, the real strain measured by the OFS being different from the one of the host structure [9]. Also, these two coatings behave differently under the effect of a static strain field. Indeed, as shown experimentally by [10], the strain measurement with a polyimide-coated fiber is closer to the true deformation in the material than with an acrylate-coated fiber. This phenomenon is due to a slippage between the acrylate coating and the cladding, which induces a loss in the strain transfer to the fiber core. More recently, this behavior was formalized by [11], who established a general solution for predicting the strain measured by a distributed OFS, whether bonded or buried, given the strain field in the host material. For non-distributed sensors, such as FBGs, similar studies have been carried out on a smaller scale; ref. [12] presents a model for quantifying the strain rate transferred to the FBG for different adhesive thicknesses and coating types in the case of embedded fibers. Later, ref. [13] investigated the influence of coating and adhesive on the strain transfer to embedded FBGs. These studies focused on the case of a static strain field. Here, we investigate the influence of the coating on dynamic strain fields measurements with FBGs, which has not been studied yet.

We propose to embed several types of optical fibers in two different polyurethane resins. Only polyimide or acrylate coating are used to protect the fibers, to avoid the influence of additional protective layers on fiber response. This allows us to use very small-diameter fibers (100–250 µm), which are less intrusive but also more fragile than jacketed fibers. First, the static strain profiles caused by the polymerization of each resin are measured using fibers without FBGs interrogated with a high spatial distributed sensing system (based on OFDR Rayleigh technology) in order to evaluate the static strain endured by the FBGs. Then, ultrasonic measurements are conducted in the resin samples. A previous study [14] was dedicated to the influence of the polymerization of two polyurethane resins on the reflectivity peak of the FBG spectra inscribed in different fiber types and on the amplitude of the ultrasonic signal recorded by these embedded FBGs. In the present work, we further focus on the shift in wavelength of the FBG reflectivity spectra after polymerization, and we evaluate whether the fiber coating can have a significant impact on the sensitivity of an FBG to the dynamic strain field caused by an ultrasonic wave.

We begin by presenting the experimental setup, which includes a description of the used specimens and measurement setups. Next, we present the results regarding the effects of embedding FBGs in a material with shrinkage. Finally, we describe ultrasonic measurements carried out on the resin samples.

## 2. Experimental Setup

In this section, we first introduce the two polyurethane resin samples and the various types of embedded optical fibers. Next, we detail the methodology for measuring the FBG reflectivity spectra. This is followed by an explanation of the approach used to determine the 1D strain profiles during the resin polymerization phase. Finally, we describe the ultrasonic measurement process, beginning with a brief introduction to the “edge filtering method” employed for measuring ultrasonic waves with FBG, and concluding with a description of the FBG interrogator and the ultrasonic measurement setup.

### 2.1. Embedding OFSs in Two Polyurethane Resin Samples

To evaluate the impact of resin shrinkage on the sensors, two cylindrical samples of polyurethane resin were prepared, each containing optical fibers embedded along their axis. The molds used to cast the resin cylinders are throughgoing to allow the optical fibers to emerge on either side. The resin was injected from one end of the mold (held vertically), while the other end was sealed to prevent any resin leakage. In this way, when the resin cured, the expansion forces were blocked, while the axial forces were free. The entire casting process took place in a thermally isolated room to reduce temperature variations. Figure 1 displays this temperature (dark curve) as well as the exothermic curves of the two resins during their polymerization (red and green curves for the F50P and the F50B24 resin, respectively). Note that it is recommended to wait between 6 and 12 h before remolding and approximately one week for near-complete polymerization. The first sample, labeled F50P, consisted of pure Sika Biresin^®^ (Zurich, Switzerland) F50 resin. The second sample, labeled F50B24, was composed of the same resin with an additional 24% mass of glass microbeads (RZ1476 filler from Breizh Baits^®^, Roubaix, France). Adding fillers to polyurethane resins is a common practice to modify their mechanical properties and modify shrinkage and exothermic reactions during the polymerization process. Each resin sample was cast in a mold with optical fibers in place. The samples are cylindrical, measuring 200 mm in length and 30 mm in diameter (Figure 2a,b).

In this study, all FBGs were microvoid gratings inscribed point by point with a 515 nm femtosecond laser. Ten optical fibers were arranged in a circular configuration with a diameter of 10 mm (Figure 2b), centered along the cylinder’s axis. The first five fibers had three FBGs each inscribed at wavelengths of 1540 nm (FBG1), 1550 nm (FBG2), and 1560 nm (FBG3) at distances 50 mm, 100 mm, and 150 mm, respectively, from one end of the sample (Figure 2a). The different fiber types are:SMF-28 fiber with an acrylate coating and a diameter of 245 μm (Fiber A in Table 1);SMF-28 fiber with a polyimide coating and a diameter of 155 μm (Fiber P in Table 1);Bend-insensitive fiber with a polyimide coating and cladding diameter of 125 μm (Fiber BIF125 in Table 1);Bend-insensitive fiber with a polyimide coating and cladding diameter of 80 μm (Fiber BIF80 in Table 1);Fiber with seven cores arranged in a hexagonal pattern with a central core, all enclosed in a 245 μm acrylate coating (Fiber 7C in Table 1). Only the central core is inscribed with three FBGs.

The remaining five optical fibers are similar but without FBGs in order to measure the 1D static strain profile via OFDR Rayleigh measurements. For this study, only the A and P fibers were used for measuring the 1D strain profiles.

### 2.2. Measurement of the FBG Reflectivity Spectra Before and After Polymerization

To evaluate the static strain experienced by the FBGs during the resin polymerization phase, we analyzed the reflectivity spectra of each FBG before and after polymerization. This analysis was performed using a BraggMETER (FS22 from HBK^®^, Darmstadt, Germany). Light from a laser swept from 1530 nm to 1570 nm was transmitted through the fiber core, and the resulting reflected spectra were recorded. This approach allowed us to measure shifts in the central wavelength and relative variations in the amplitude of the FBG reflectivity spectra induced by the polymerization of the resin. Note that these measurements were made in an environment where the temperature was controlled.

### 2.3. 1D Static Strain Profiles Obtained by OFDR Rayleigh Measurement

Additionally, the 1D strain profiles along the axis of the cylindrical samples were obtained by interrogating the A and P fibers without FBGs (see Table 1) using the OFDR Rayleigh method [15]. Measurements were performed with a ODiSI-B system from Luna^®^ (Blacksburg, VA, USA). The gauge length was set to 1.3 mm, with a spacing of 0.65 mm between consecutive gauge lengths, and a sampling frequency of 23.8 Hz.

### 2.4. Ultrasound Measurement Chain

The FBG interrogator system developed by the CEA includes an external cavity tunable laser with wavelength range 1527–1565 nm manufactured by Yenista Optics^®^ (Lannion, France). The laser intensity is adjusted using variable optical attenuators. The light backpropagated by the FBG is transferred, via a circulator box, to a photodiode card to convert the optical signal into an electrical one. The direct current part of the resulting analog signal is sampled at a low acquisition rate (≈2 Hz) to display the FBG reflectivity spectra. The edge filtering method [16] was also employed. This method involves adjusting the wavelength of the laser source to one of the edges (assumed to be linear) of the reflectivity spectra of the FBG being interrogated. In this way, a shift in the central wavelength of the spectra, due to deformation of the medium surrounding the FBG, is perceived as a variation in reflectivity by the laser source and thus interpreted as the propagation of an ultrasonic wave. The alternative current part is band-pass filtered between frequencies 30 kHz and 250 kHz before being sampled at a high sampling rate (10 MHz). The ultrasonic dynamic deformation perceived by FBG1, FBG2, and FBG3 (Figure 3a) were recorded and displayed using an oscilloscope. In this study, no multiplexing was used; hence, we made the choice to interrogate two FBGs at the same time, each one inscribed in a distinct fiber plugged into two different circulator boxes.

The ultrasonic source is a dry-contact piezoelectric transducer (longitudinal Acsys S1803) positioned at one end of the resin sample (Figure 3b) and emits a 5 μs rectangular pulse. The resulting wave has a central period of 2 × 5 μs, so a central frequency of 100 kHz.

## 3. Study of the 1D Strain Profiles of the Polyurethane Resins and Quantification of the Effects on the Central Wavelength of the FBG Reflectivity Spectra

In this section, we present the results concerning the impact of the insertion of FBGs into a material subject to shrinkage on the reflectivity spectra of several FBGs.

Embedding FBGs in a castable resin can alter their reflectivity spectra during polymerization, potentially leading to a shift in the central wavelength, changes in the shape of the reflectivity peak, and a reduction in the amplitude of the reflectivity peak due to optical attenuation. It is therefore crucial to control the shape, wavelength, and amplitude of FBG reflectivity spectra to correctly configure the edge filtering method, prior to performing acoustic measurements. Figure 4 shows the reflectivity spectra of FBG1, FBG2, and FBG3 inscribed in the fibers A and P, and embedded in resins F50B24 (Figure 4a) and F50P (Figure 4b), both before (solid lines) and after (dotted lines) polymerization. These figures highlight two major effects of the shrinkage forces during resin polymerization. First, amplitude variations of the reflectivity peak of each FBG are observed. In some cases, the amplitude decreases so drastically that the reflectivity peaks become undetectable after polymerization (see Figure 4b). In other cases, these variations are smaller and even show an increase in amplitude for some spectra (see Figure 4a). The origin of these amplitude variations is discussed in greater detail in the following study [14]. Second, a shift in the central wavelength ($\lambda_{B}$) of the reflectivity spectra is observed for all FBGs (Figure 4). This shift results from shrinkage forces compressing the FBGs axially, leading to a reduction in their central wavelengths. We believe that in the present case, axial shrinkage may have been amplified by the elongated shape of the mold, which allowed shrinkage only in the direction of the fibers.

Figure 5 illustrates the wavelength shifts ($\Delta \lambda_{B}$) of the reflectivity peaks for FBGs embedded in the tested resins. For the F50P resin (Figure 5a), only the $\Delta \lambda_{B}$ values of the BIF125, BIF80, and 7C fibers are displayed, as the reflectivity spectra of the FBGs inscribed in the A and P fibers were either severely attenuated or entirely lost (see Figure 4b). We observed that the type of resin significantly influences the results. Indeed, the average $\Delta \lambda_{B}$ values are 3–4 times less intense for resin F50B24 (Figure 5b) than for F50P (Figure 5a). Nevertheless, the wavelength shift for FBG1 inscribed on the P fiber and embedded in the F50P resin (see Figure 4b) is much smaller (≈−2.5 nm) than the other wavelength shifts observed for this resin. This difference may be due to slippage of the P fiber during curing of the F50P resin, such that FBG1 would have been at a position where the deformation was less intense than expected (i.e., a smaller wavelength shift than expected) and therefore would have been less affected by shrinkage than FBG2 and FBG3. Beyond that, the observations made in Figure 5 are in good agreement with the 1D strain profiles measured along the axis of the resin samples using the A and P fibers without FBGs (Figure 6). First, we note that the difference that exist between acrylate-coated fibers (solid curves) and polyimide-coated fibers (dotted curves) for static strain gradients is clear for the F50B24 resin (Figure 6b). For the F50P resin (Figure 6a), only the 1D strain profile measured by the A fiber is plotted because some complications during the measurement campaign prevented us from processing the data of the P fiber (we suppose that the fiber is broken). Moreover, the fact that the 1D strain profile shown for the A fiber is incomplete could arise from an excessive strain regime applied by the F50P resin during polymerization, which may have exceeded the operating range of the sensing system (see Section 2.3). Consequently, we decided to exclude this resin from the ultrasonic measurement campaign presented in the next part. Second, these graphs demonstrate that the deformations induced by polymerization are significantly greater in the F50P resin, with maximum strain (below −7000 µm/m) more than twice that observed in the F50B24 resin (≈−3000 µm/m). This explains the stronger shifts and amplitude losses observed for F50P, as shown in Figure 4. Furthermore, the strain profile in the F50B24 resin reveals that the shrinkage forces during polymerization are more intense near the ends of the samples (≈−3000 µm/m) compared to the center (≈−1400 µm/m) and thus emphasizes the importance of understanding the spatial stress distribution within the material in optimizing the placement of the sensor.

Additionally, we have compared the local strain measured by the FBGs embedded in the F50B24 resin with the strain profile measured with the high spatial distributed sensing system. These local strain measurements were determined using the following equation [17]:(1)$$\epsilon =\frac{\Delta \lambda_{B}}{\lambda_{B}}\times \frac{1}{\left(1-p_{e}\right)},$$
where $p_{e}$ is the photo-elastic constant of the optical fiber core (≈0.22 for a silica core [18]), ϵ is the strain measured by the FBG, $\lambda_{B}$ is the initial Bragg wavelength, and $\Delta \lambda_{B}$ is the difference between $\lambda_{B}$ before and after the polymerization of the resin. For the $\Delta \lambda_{B}$ value, we used the average $\Delta \lambda_{B}$ observed for the FBGs inscribed in the A and P fibers. The comparison is visible in Figure 6b and show a good agreement between the 1D strain profiles and these local strain values (red stars for the A fiber and blue stars for the P fiber). Thus, we can conclude that with Bragg gratings inscribed in these types of optical fiber embedded in resin, we are able to obtain a good estimate of the local deformation of the material present around the grating during its polymerization.

Finally, all these results highlight the importance of understanding the strain regime applied by the hardening of the host material in order to anticipate the effects on the FBG reflectivity spectra. This is particularly important when the wavelength of the FBG reflectivity peak is expected to match the wavelength of the interrogation laser.

## 4. Study of the Influence of the Acrylate and the Polyimide Coating on the Measurement of Ultrasonic Waves with Embedded FBGs

In this section, we focus on comparing the signal amplitude, waveform, and noise level between acrylate and polyimide coatings. In particular, we compare the energy of an ultrasonic wave measured by an acrylate-coated FBG and a polyimide-coated FBG.

Figure 7 shows 35 pre-stack ultrasonic signals (over 105) recorded by the three FBGs inscribed in the A and P fibers embedded in the F50B24 resin sample. Note that each pre-stack is the result of decoupling and recoupling the ultrasonic source on the resin sample to test measurement repeatability. Based on the positions of the sensors in the resin sample (Figure 2a) and the P-wave velocity in this resin (around 2100 m/s, velocity measured by a first-arrival time peaking between a piezoelectric source and a vibrometer positioned on the ends of another F50B24 resin sample), the theoretical travel times are as follows: tt1 = 71 μs for FBG1, tt2 = 48 μs for FBG2, and tt3 = 24 μs for FBG3. These expected travel times are represented by the red lines in Figure 7. There is good agreement between the theoretical and experimentally observed travel times. However, for FBG2 and FBG1, early arrivals are observed before the expected arrival times of the P-wave (highlighted by purple rectangles in Figure 7). The travel times of these premature arrivals match perfectly with the first arrivals from the previous FBGs, suggesting a possible crosstalk effect between the three FBGs of the same fiber. This phenomenon probably also pollutes the rest of the signal from each FBG; however, this analysis is complex because various wave packets, corresponding to reflections on the interfaces of the medium, may occur at these later times. Note that the FBGs have different center wavelengths, which is a configuration in which no crosstalk due to a Fabry–Pérot effect is expected. The origin of this observation is discussed in the next section. Moreover, the sign and amplitude of the presumed crosstalk effect changes from one signal to another, which means that the stacking process could decrease this phenomenon by summing.

The stacked signals are presented in Figure 8, with the theoretical travel times for each FBG (red lines) and the purple rectangles, which highlight the presumed crosstalk between FBGs on the same fiber. Note that each signal displayed in this figure is the result of the stacking of the repeated measurements for each FBG independently. Overall, these signals show no significant differences in amplitude or noise level between the two coatings. Regarding the waveform, the signals acquired by FBG1 are very similar, with noticeable differences starting from t1 = 125 μs (Figure 8), probably due to the difference in position between the two optical fibers. As explained in the previous paragraph, we see that the presumed crosstalk effect observed in the pre-stack data (Figure 7) is minimized through the stacking process for this FBG. In contrast, the crosstalk effect remains noticeable in the signal recorded by FBG2 even after the stacking process. For the signals acquired by FBG3, the waveforms of the first arrivals are identical, with noticeable differences emerging at t3 = 50 μs (see Figure 8), probably for the same reason explained for FBG1.

Finally, the first arrivals show a good fit with the predicted travel times and a good repeatability of the waveform across different coatings, despite the presumed crosstalk observed between the FBGs on the same fiber. By contrast, after the direct paths, there are differences in amplitude and waveform for the secondary arrivals. This is probably due to the difference in position between the two fibers in the resin sample (Figure 2b). These results lead to the conclusion that the coating type does not significantly affect the signal amplitude or noise level, especially for the direct paths.

In addition to the previous results, several pairs of signals were recorded using the same setup for all three FBGs but with varying source intensities. The goal was to compare the energy of the dynamic strain signal transferred to the fiber core, where FBGs are inscribed, through the acrylate and polyimide coatings. The results are presented in Figure 9, which uses logarithmic scales. The plot shows the energy of the stacked signals recorded by the FBG for the acrylate-coated fiber on the y-axis and the polyimide-coated fiber on the x-axis. The energy of a signal is calculated as follows:(2)$$E=\sum_{i=1}^{n_{ech}}{\left|x_{i}\right|}^{2}.\delta t,$$
where δt is the sampling rate equal to $10^{-7}$ s, and $n_{ech}$ and $x_{i}$ represent the number of points and the *i*th sample of *x*, respectively. Three curves are shown in this figure, green, purple, and blue, corresponding to the energy of the signal measured by FBG1, FBG2, and FBG3 inscribed in the A and P fibers. Each curve contains seven distinct points, representing seven different levels of source amplitude ([0.803, 1.033, 1.297, 1.628, 2.029, 2.558, 3.209] V) for the ultrasonic source. As expected, the higher the source intensity, the greater the energy transmitted to the FBG. Additionally, since FBG1 is the measurement point furthest from the source, its energy values (calculated using Equation (Equation 2)) are lower than those of FBG2 and FBG3 for a given source intensity ([0.55 × 10^−3^; 1.2 × 10^−2^] V^2^·s vs. [2.8 × 10^−3^; 7 × 10^−2^] V^2^·s). Despite the spacing between FBG2 and FBG3, the energy values transmitted to the fiber core are quite similar for both FBGs, probably partly due to crosstalk, which increases the energy measured by FBG2. Finally, it can be observed that, regardless of the coating type used, the energy transferred to the fiber core is similar. Therefore, the difference in behavior observed between acrylate and polyimide coatings for a static strain gradient (see Figure 6b) does not hold for an embedded FBG subject to a dynamic strain gradient caused by an ultrasonic wave.

## 5. Discussion

Firstly, using two test samples made from polyurethane resins, we were able to quantify the shift in the central wavelength of the reflectivity peaks of several FBGs inscribed in different types of optical fibers, as influenced by the polymerization phase of the resins. This study revealed that, depending on the type of resin, the value of the shift $\Delta \lambda_{B}$ can reach almost −10 nm, which may prevent us from interrogating the FBGs depending on the suitability of the laser source used. For instance, in the case of an FBG embedded in resin with a strong shrinkage regime, if the Bragg wavelength is not properly accounted for in relation to the shrinkage imposed by the material, the FBG spectra could shift outside the acceptable spectral range of the optoelectronic system, making the sensor unusable.

Secondly, ultrasonic measurements were conducted on the F50B24 resin sample to investigate whether the type of coating could significantly affect the amplitude, noise level, and waveform of the recorded ultrasonic signals. As shown in Figure 8, despite the presumed crosstalk depicted between the FBGs of the same fiber, there is good agreement between the theoretical and experimental travel times of the first arrivals, with no significant differences in amplitude or noise level between the two coatings. The waveform of the first arrivals also appeared to be repeatable. Furthermore, the energy of the ultrasonic signal transferred from the host material to the fiber core was examined (see Figure 9). Contrary to the static strain profiles resulting from the resin polymerization, where acrylate and polyimide coatings exhibit different behaviors, no significant differences could be observed between the two coating types in terms of energy transmission to the fiber core. Ultimately, these ultrasonic measurements lead us to conclude that the difference in behavior observed between acrylate and polyimide coatings under the static strain induced by polymerization (see Figure 6b) does not hold for the response to the dynamic strain caused by ultrasonic waves on the embedded FBG.

Additionally, the ultrasonic measurements demonstrated the possibility of using FBGs embedded in polyurethane resin to measure ultrasonic wave signatures with repeatable direct path waveforms and with travel times validated by theoretical predictions. Moreover, optical fibers, with their low intrusiveness (100–250 µm, with only one protective coating), are excellent candidates as mechanical wave sensors for integration into structures with a limited effect on their mechanical properties. Further tests regarding the mechanical behavior of such structures are still needed, in particular regarding their fatigue behavior. As shown by [19], the use of reduced-scale physical modeling, through mock-ups made from resin, is an effective way to validate geophysical methods before applying them in the field. Epoxy and polyurethane resins, when stored in temperature-controlled environments, are highly reproducible and temporally stable materials. They also have P-wave and S-wave velocities ($V_{p}$ and $V_{s}$) ranging from 2000 to 3000 m/s and 800 to 1500 m/s, respectively, which are orders of magnitude higher than those often found in the field. Consequently, the use of these materials is of major interest in reduced-scale modeling (a few centimeters to few dozen centimeters) of large-scale near-surface seismic wave propagation (a few meters to few dozen meters). Therefore, this study supports the use of FBGs embedded in a polyurethane resin mock-up, inscribed in a low-intrusiveness optical fiber, as ultrasonic wave sensors to model a full-size experiment and potentially validate imaging methods.

Furthermore, a crosstalk phenomenon may explain the early arrivals observed in the ultrasonic data. One possible explanation is the formation of a Fabry–Pérot cavity between the Bragg gratings of a fiber. This is a well-known phenomenon in which two FBGs with an identical wavelength are inscribed in a fiber (see [20]). Here, the phenomenon is unexpected: as mentioned in Section 2.1 the different FBGs of a fiber are inscribed at different Bragg wavelengths. Despite this fact, it is possible that the residual reflection of a grating outside its reflection peak induces interference with the reflected light from the interrogated grating. This would lead to interference peaks, superimposed with the Bragg reflection peaks but spectrally much narrower. See [21] for an example of ultrasonic wave measurement with an unstabilized fiber Fabry–Pérot interferometer (FFPI) in a setup quite similar to ours (embedded fiber, ultrasonic wave emission with a PZT transducer). Hence, the early arrival signals observed when interrogating FBG2 and FBG1 may come from the following mechanism: the US waves first impact the grating FBG3, not interrogated, at time of flight tt3, as shown in Figure 8. This does not lead to a significant change in its power reflection coefficient at the laser wavelength. Indeed, the laser is operating 10 nm away from the FBG3 Bragg wavelength. However, the Fabry–Pérot cavity length between FBG2 and FBG3 changes slightly due to the compression and expansion of FBG3. This leads to a change in the interference peak’s spectral period. Hence, at the laser wavelength, at time of flight tt3, a change in the reflection coefficient can be detected despite the fact that the FBG3 is not interrogated. Of course, the same explanation would hold at time of flight tt2 in the FBG1 signal. A possible way to clarify this phenomenon would be to embed another optical fiber with only one FBG placed at the same position as FBG2 or FBG1 of the other optical fiber and determine whether we observe the same phenomenon on this FBG. Further experiments will be conducted to investigate this hypothesis in detail and provide a quantitative explanation of the phenomenon.

In this contribution, we have gained a better understanding and control of the challenges involved in embedding FBG sensors in castable materials subject to strong static strains during their polymerization phase, like polyurethane resin. We have realized the importance of knowing in advance the material into which optical fibers are embedded for two key reasons. First, by having a quantitative estimation of the internal strain regime applied by the material during its polymerization phase, we can inscribe the Bragg gratings at a wavelength higher than that required by the interrogation system to anticipate the shift in the central wavelength of the FBG spectra caused by material shrinkage. Second, knowing the spatial distribution of strains along the fiber is helpful in identifying areas of high shrinkage intensity and avoiding embedding FBGs in these critical regions. Additionally, the ultrasonic measurement campaign demonstrates the possibility of measuring ultrasonic signals with FBGs embedded in polyurethane resin and also theoretically validating the first-arrival times between the source and the various embedded Bragg gratings. Furthermore, analysis of the ultrasonic signals revealed a potential inter-FBG crosstalk effect. This phenomenon was unexpected, as the FBGs are inscribed at different wavelengths.

## 6. Conclusions

This study presents an experiment designed to investigate two phenomena. First, it examines the impact of the polymerization phase of two different polyurethane resins on the reflectivity spectra of embedded FBGs. Second, it compares the influence of acrylate and polyimide protective coatings on optical fibers embedded in the same polyurethane resins, focusing on the transfer of energy from an ultrasonic wave in the host structure to the fiber core. To achieve this, various types of optical fibers were embedded in two different polyurethane resins (F50P and F50B24), including some with FBGs and others without. The main findings of this study are as follows:In the case of a material with a pronounced shrinkage regime (as observed in the F50P resin), the resulting wavelength shift can be significant enough to cause the FBG reflectivity spectra to shift by up to 10 nm. Therefore, it is crucial to account for the central wavelength shift of the Bragg gratings caused by material shrinkage to ensure that the reflectivity spectra remains within the accessible spectral range of the interrogator.Under the effect of a dynamic strain field, the type of protective coating on the optical fiber (acrylate or polyimide) was observed to have a minimal influence on the energy of the ultrasonic signal transmitted from the host material to the fiber core.

For materials with important shrinkage (up to −7000 µm/m), a single acrylate or polyimide protective layer was insufficient to adequately protect the FBG sensors. Therefore, one potential direction for future research would be to explore whether these findings can be extended to more robust protective coatings. Additionally, the comparative study of the effect of the fiber’s protective coating on the energy of an ultrasonic wave transmitted from the structure to the fiber core was carried out for an ultrasonic wave with a center frequency of 100 kHz and a bandwidth from 20 kHz to 250 kHz. Extending this comparative study to a higher frequency regime could reveal a difference in behavior between the two coatings under the effect of an ultrasonic wave, and it could also be useful to learn more about the frequency response of the sensors used. Finally, although this work has highlighted certain difficulties associated with burying optical fibers in a castable material, we have shown that it is possible to obtain usable acoustic measurements. In particular, we have observed that it is possible to measure consistent P-wave travel times between the source and a buried FBG. Our future work will focus on three axes: first, the possibility of exploiting these waves to probe a medium, for example, by applying a tomography method based on P-wave travel times between several sources and several receivers in a resin block; second, the detailed investigation of the hypothesis of crosstalk between two FBG inscribed at different wavelengths and providing a quantitative explanation of the phenomenon; third, extending the comparative study of the two coatings to study the influence of increasing the frequency of the ultrasonic wave on these coating behaviors.

## Figures and Tables

**Figure 1 sensors-25-02657-f001:**
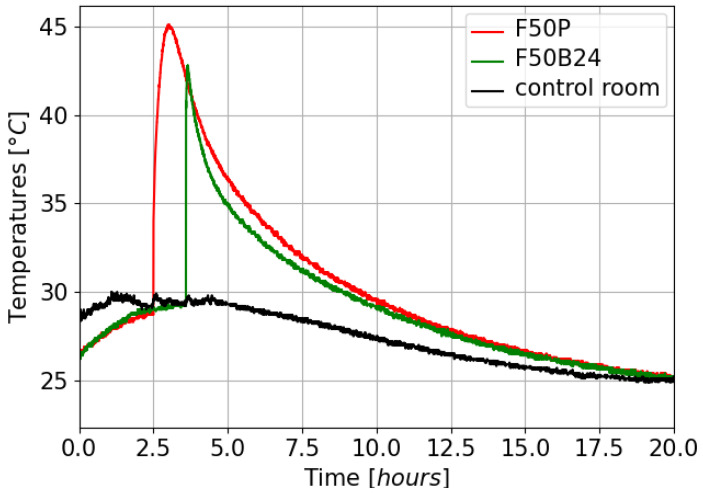
Temperature of the control room, as well as exothermic curves of the two resin during their polymerization.

**Figure 2 sensors-25-02657-f002:**
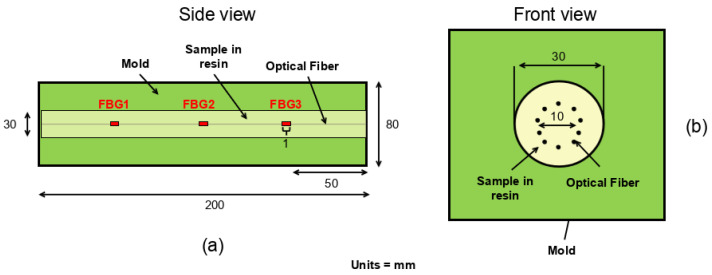
Conceptual schemes showing the embedded FBGs and the geometry of the resin samples. (**a**) Schematic longitudinal section of the mold showing the position of the FBGs along each inscribed fiber. FBG1, FBG2, and FBG3 denote the three FBGs of one fiber. (**b**) Front view of the mold with the resin sample inside.

**Figure 3 sensors-25-02657-f003:**
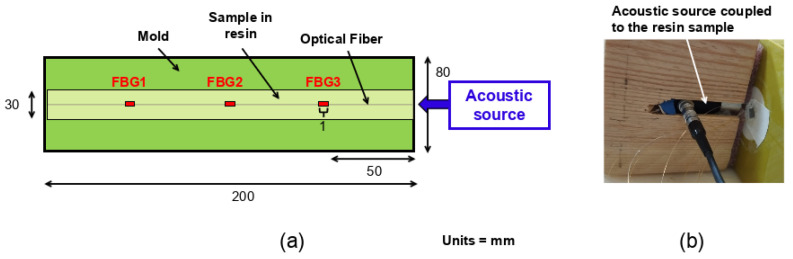
Conceptual scheme of the experimental setup used to measure a dynamic strain field with FBGs embedded in the resin samples. (**a**) Conceptual scheme of the sample, showing the position of FBGs and the acoustic source. FBG1, FBG2, and FBG3 denote the three FBGs of one optical fiber. (**b**) Photograph of the acoustic source coupled with the F50B24 resin sample.

**Figure 4 sensors-25-02657-f004:**
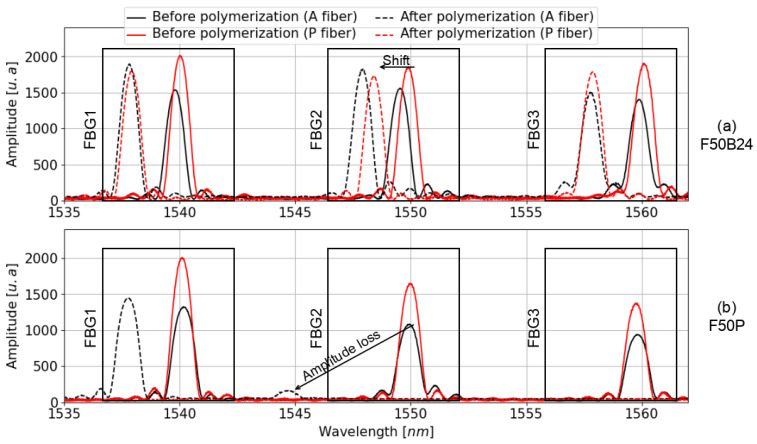
Reflectivity spectra of the three FBGs inscribed on the A (in black) and P (in red) fibers before (solid lines) and after (dotted lines) the polymerization of the F50B24 (**a**) and F50P (**b**) resins. The black boxes delimit the location of the FBG1, FBG2, and FBG3 spectra. The arrows labeled “shift” and “amplitude loss” highlight the main effects attributed to the shrinkage of the resin during the polymerization.

**Figure 5 sensors-25-02657-f005:**
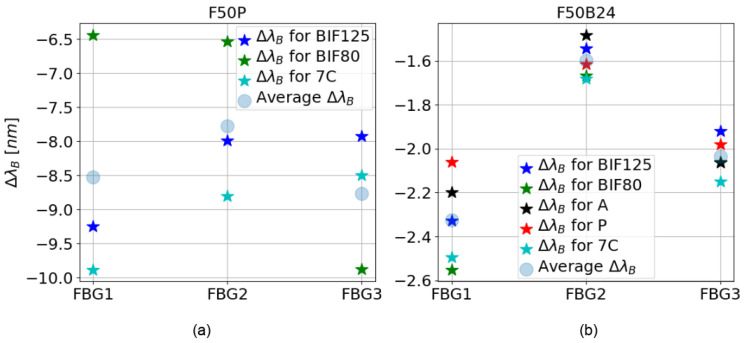
Shift $\Delta \lambda_{B}$ of the central wavelength induced by the material shrinkage of the three FBGs of each inscribed fiber type embedded in the two resin samples. (**a**,**b**) represent the shift of the central wavelength for the optical fibers embedded in the resin F50P and F50B24, respectively.

**Figure 6 sensors-25-02657-f006:**
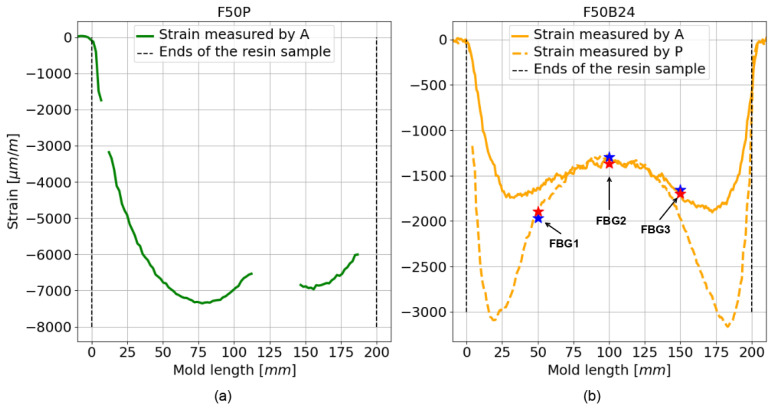
1D static strain profile along the axes of the resin samples. (**a**,**b**) represent the 1D static strain profile for the optical fibers embedded in the resin F50P and F50B24, respectively. The blue and red stars correspond to the local strain values measured using Equation (Equation 1) and the average wavelength shift of the FBGs inscribed in the P and A fiber, respectively.

**Figure 7 sensors-25-02657-f007:**
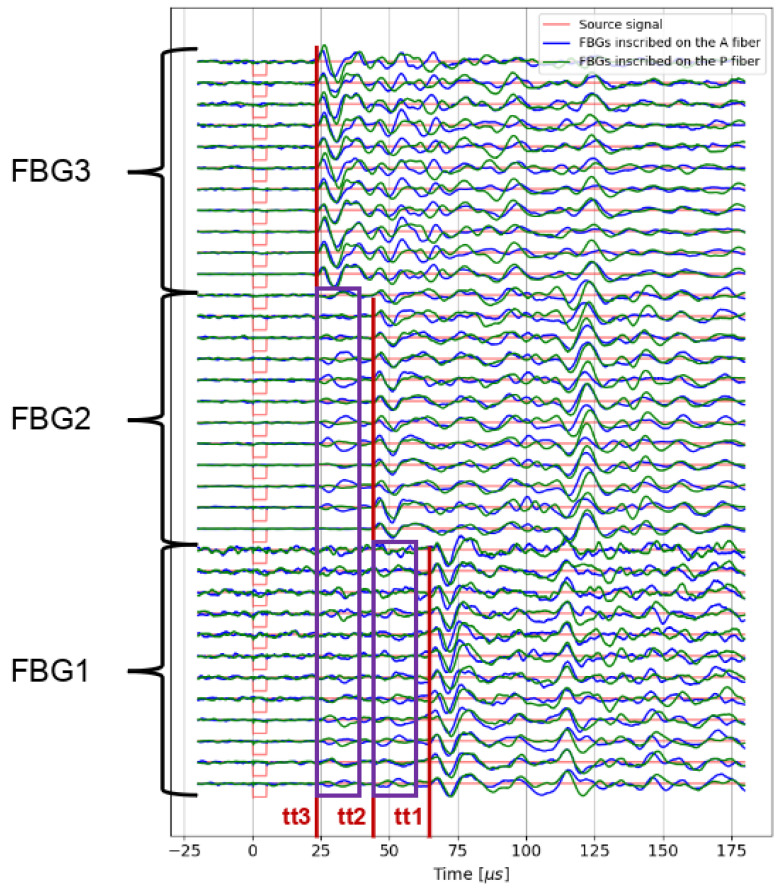
Pre-stack data recorded by the three FBGs inscribed in the A fiber (green curves) and P fibers (blue curves) embedded in the F50B24 resin sample. The red lines represent the predicted travel times, tt1, tt2, and tt3, for the three FBGs. The purple rectangles represent the crosstalk observed between the three FBGs.

**Figure 8 sensors-25-02657-f008:**
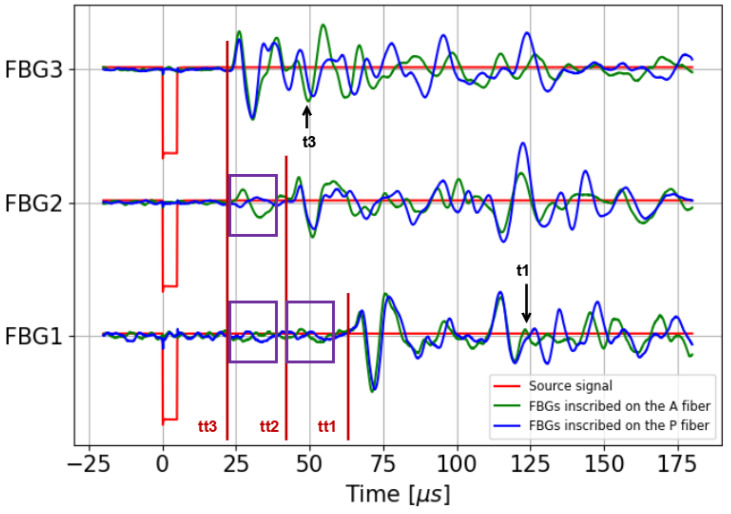
Stacked ultrasonic signals recorded by the three FBGs inscribed in the A (blue curve) and P (green curve) fibers inside the F50B24 sample. The red curve correspond to the source signal.

**Figure 9 sensors-25-02657-f009:**
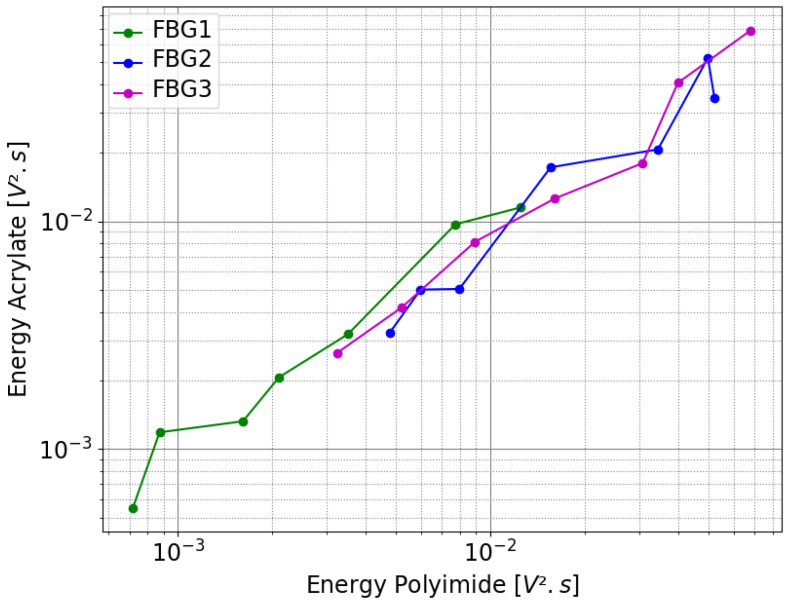
Comparison of the energy of the signal transmitted to the FBG between an acrylate-coated fiber and a polyimide-coated fiber for various source intensities.

**Table 1 sensors-25-02657-t001:** Names and types of optical fibers embedded in the two resin samples. The coating diameter indicates the total diameter of the fibers.

Fiber Name	Fiber Type	Cladding Diameter (µm)	Coating Diameter (µm)	Coating Type	FBG Size (mm)
A	Standard Telecom Single Mode Fiber (SMF-28)	125	245	Acrylate	1
P	Standard Telecom Single Mode Fiber (SMF-28)	125	155	Polyimide	1
BIF125	Bend-Insensitive fiber (BIF), single mode	125	155	Polyimide	1
BIF80	Bend-Insensitive fiber (BIF), single mode	80	155	Polyimide	1
7C	7 cores fiber, single mode	125	245	Acrylate	1

## Data Availability

All the data required to reproduce the results of this article are available at https://doi.org/10.57745/9IYNUO.

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
