# Peer review of "Challenges of Embedding Fiber Bragg Grating Sensors in Castable Material: Influence of Material Shrinkage and Fiber Coatings on Ultrasonic Measurements"

_sensors, 2025, doi:10.3390/s25092657_

Round 1

Reviewer 1 Report

Comments and Suggestions for Authors

This work embedded FBG sensors in polyurethane resins and measured the strain changes during the polymerization process. The paper also introduced the sensitivity of FBGs to ultrasonic waves and compared the results with different coatings. The paper seems to be original and well organized. The paper can be considered for publication after the revisions mentioned below:

  1. What are the parameters of the laser? What is the loss of each FBG, and what is the purpose of using microvoid FBG instead of regular FS laser induced FBG?
  2. In section 2.4, the authors mention that “the optical setup was configured to interrogate two FBGs, each one on a distinct fiber.” Could the authors explain this method in detail?
  3. In lines 176-178, why is shrinkage allowed only in the fiber direction? Why is axial contraction amplified by the mold shape? I thought the mold would only prevent expansion and not shrinkage.
  4. Why are fiber A and P not as effective as the other three in strain measurement?
  5. The diameter of the fiber circle is only 10mm, and the resin is 30mm. How can it slip out? Is the fiber straight before solidification? Is there any pretension?
  6. What is the strain sensitivity of the FBG sensors with different coatings?
  7. The strain data of fiber A in Figure 5 (a) is incomplete, there is no data for fiber P, and no comparison with FBG is included. Is the data in Figure 5 (a) accurate? Can the author add simulation results for comparison?
  8. What is the purpose of stacking the signals from three locations in the ultrasound experiment?
  9. What are the values of the 7 source intensities?
  10. Which coating is better in the polymerization process, Acrylate or Polyimide?
  11. In the crosstalk explanation, I am still confused about how a small change in the FBG spacing affects the FBG's reflection coefficient.

Reviewer 2 Report

Comments and Suggestions for Authors

In this work, the authors investigated the feasibility of embedding fiber Bragg gratings (FBGs) in polyurethane resins for ultrasonic measurement. The effect of material shrinkage and fiber coating on static/dynamic strain measurement is discussed, which has clear engineering application value. In my opinion, this manuscript should be considered after addressing the following issues.

  1. The curing conditions of the two resin materials, such as curing temperature and time, need to be clearly given in the manuscript.
  2. For Fig. 6, authors said “... for FBG2 and FBG1, early arrivals are observed before the expected arrival times of the P-wave...” and “The travel times of these premature arrivals match perfectly with the first arrivals from the previous FBGs, suggesting a possible cross-talk effect between the three FBGs of the same fiber.” In my opinion, this explanation is lacking in evidence. The cause of crosstalk should be given in detail. The reflection spectra of fiber A and fiber P used in this experiment should be given in the manuscript for evaluating whether FP interference will be generated during testing. In addition, the crosstalk can be proved by placing a fiber with only one FBG in the same position as FBG1 or FBG2 in the polymerization resin.
  3. In the conclusion, authors said “Under the effect of a dynamic strain field, the type of protective coating on the optical fiber (acrylate or polyimide) was observed to have a minimal influence on the energy of the ultrasonic signal transmitted from the host material to the fiber core.” Only one ultrasonic frequency is considered in this work. Will the two coating materials have different effects on the transmission of ultrasonic signals at higher frequencies?

Round 2

Reviewer 1 Report

Comments and Suggestions for Authors

No major issue detected. The paper can be considered for publication.

Reviewer 2 Report

Comments and Suggestions for Authors

This manuscript can be accepted in present form.